# Mitigation bank applications for freshwater systems: Control mechanisms, project complexity, and caveats

**Sebastian Theis**[ORCID]*, **Mark Poesch**

Fisheries and Aquatic Conservation Lab, Faculty of Agricultural, Life and Environmental Sciences, University of Alberta, Edmonton, Canada

* theis@ualberta.ca

## Abstract

Biodiversity and mitigation banking has become a popular alternative offsetting mechanism, especially for freshwater species and systems. Central to this increase in popularity is the need for sound control mechanisms to ensure offset functionality. Two commonly used mechanisms are monitoring requirements and staggered release of bank credits over time. We used data from 47 banks in the United States, targeting freshwater systems and species. Based on the 47 banks meeting our criteria we showed that control mechanisms generally scale with increased project complexity and that banks release most of their total credit amount within the first 3 years. We further showed that advance credits are common and can increase the potential for credit release without providing tangible ecological benefits. Physical and biological assessment criteria commonly used by banks let us identify three main bank types focusing on connectivity, physical aspects, and habitat and species and their application possibilities and caveats to provide different ecosystem benefits for freshwater species and systems affected by anthropogenic development.

## 1. Introduction

Offsetting is a legal requirement in many countries that allows for authorized serious harm to fish or fish habitat through the implementation of appropriate restoration or mitigation measures [1–4]. This requirement is becoming increasingly important due to the rapid global decline in aquatic biodiversity, especially in freshwater ecosystems while human development increases [5, 6]. Proponents are responsible for offsetting impacts to avoid serious harm to fisheries productivity and fish habitat [1, 2]. Offsetting is the last step in the mitigation hierarchy, which begins with defining the predicted impact (i.e., harm/damage) to the ecosystem, followed by mitigation measures such as avoidance of impacts by moving the project to less impacted areas, minimization through mitigation measures (e.g., staging vehicles off-site), rehabilitation (e.g., repairing riparian areas from trampling), and leaving the residual impact that needs to be offset (Fig 1; [3, 7, 8]). The goal of offsetting is to compensate for the authorized loss of species, habitat, or ecosystem services with the objective of achieving No Net Loss

**Data Availability Statement:** Data and additional material are available through contacting the authors, freely available on RIBITS and in the supplemental material or through figshare: 10.6084/m9.figshare.22864019.

**Funding:** Funding for this project was provided by Mitacs Cluster Accelerate (RES0027784) and Converge (RES0021639) grants to M.P. (https://www.mitacs.ca/en/programs/accelerate) The funders had no role in study design, data collection and analysis, decision to publish, or preparation of the manuscript.

(NNL), which means providing equal or greater benefits (net positive gains) through the compensatory mitigation measures than the residual negative impact [4, 9].

Habitat banking, such as mitigation banking, is a specialized form of traditional offsetting that has gained significant traction in the conservation of freshwater systems and species over the past few decades. While initially spearheaded and utilized mainly in the United States and Australia, habitat banking is increasingly being developed and implemented in other countries [1, 10, 12, 13]. Habitat banking follows the same mitigation methods and hierarchy as regular offsetting practices dictate, with some key differences. In habitat banking, third parties establish, implement, and manage created, enhanced, restored, or preserved habitat areas or ecosystem functions. The project comprises the bank site(s), banking agreement (legal obligations, project layout, and control mechanisms), and service area (the area where the bank can sell their credits; [1, 10, 12, 14]). Proponents purchase a required credit amount from the bank for their expected impacts, corresponding to an equal species, ecosystem service, or habitat value set by the responding agency or government body ([1, 15]; Fig 1).

To be approved by the responsible regulatory agency, a habitat bank needs to have a banking agreement and necessary permits, such as a land disturbance permit for conducting habitat

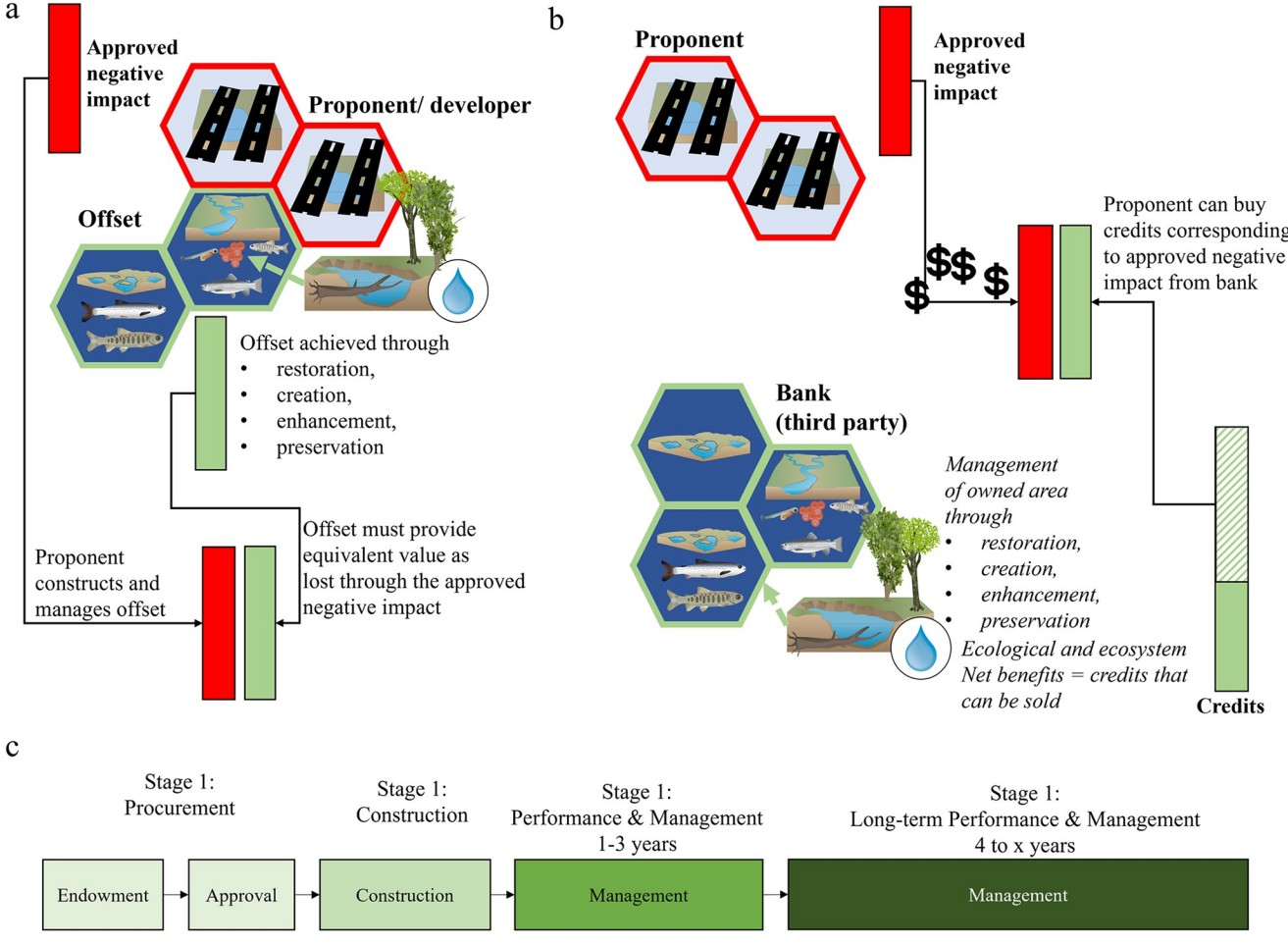

**Fig 1. Habitat banking framework.** Habitat banking is an alternative offsetting mechanism (a), with offsets provided through a third party in the form of credits bought by the proponent (b). Common stages of bank establishment to long-term management (c). (Based on [1, 3, 9, 10]; image attribution: [11]).

restoration. The banking agreement includes all bank details, including financing, sponsors, mitigation methods, and monitoring [1, 15, 16]. For this paper, "banks" will refer to the sum of the site, credits, agreement, and service area, unless otherwise specified.

Credits are generated by an Interagency Review Team (IRT) based on the banked area and the applied mitigation method (e.g., restoration, enhancement; Fig 1). Credit amounts vary due to differences in environmental benchmarks against reference sites, also called net benefit (e.g., water quality, ecological aspects, soil, vegetation; [1, 12, 15, 17]). Factors affecting credit calculations include the target system and approach, mitigation timing, monitoring, contingency plans, and control measures. Credits follow a release schedule based on the bank achieving certain benchmarks, such as the implementation of planned in-stream modification or meeting success criteria in the following years [1, 18–20]. Common monitoring approaches include indices of biotic integrity, well-being, or physical habitat stability and suitability [16–18].

Reviewing mitigation and offsetting literature, key terms that are normally used are 'Biodiversity offset' and 'Mitigation bank', which shows that the overarching use of offsetting and related studies is still new within the scientific community [21]. Its steady increase in coverage and popularity over the past 20 years (2002–2022; Fig 2A and 2B) showcases its growing importance. Looking at key studies yields three important questions that arise regarding the 'Why', 'How', and What? (Fig 2C; [3, 4, 7, 8, 22–25]).

'Why' refers to why offsetting and mitigation are necessary. Consensus on this topic is clear, with many studies referring to the rapid loss of biodiversity and habitat due to anthropogenic impacts as well as overarching background processes and the need to find a sustainable solution to allow resource use while preserving pristine habitat and compensate for authorized losses [3, 4, 21, 23, 26–28]. 'How' commonly covers the different approaches that are being used to achieve no net loss or net benefits on an environmental and ecological level. The 'How' is often focussed on the on-the-ground logistics like restoring a specific habitat or improving the productivity of a specific target species. It is grounded in decades of restoration work for both terrestrial and aquatic systems and builds on that extensive available knowledge [24, 29–34]. The 'What' leads to the point of what the outcomes of offsets are and if they ultimately meet the goals of compensating for approached negative anthropogenic impacts. Based on current large-scale review studies assessing banking and offsetting frameworks the majority of said studies focus on the environmental and ecological outcomes of offsets, the 'What' [8, 18, 22, 23, 25, 35–37]. Most studies conclude that offsets, including banks, often do not meet No net loss requirements, experience long-term failure, and that provided benefits do not adequately compensate for the initial negative impacts they are meant to offset [22, 23, 25, 29, 35, 38–40].

Gaps on the other hand, especially exist within the context of the 'How'. These gaps are not about the physical aspect of applied restoration measures, though still associated with many uncertainties, but rather concern the control mechanisms that are available to ensure project performance [8, 14, 26, 38]. This applies especially to mitigation banks where most studies are rather focused on whether banks achieve area or ecological equivalency [12, 20, 23, 35, 37]. Control mechanisms can vary across banks and systems, but two common ones are required monitoring timeframes and release schedules, where release schedules refer to the amount of time in which bank-generated credits are released and become available to proponents as described earlier (Fig 1B; [1, 15, 42]). According to literature reviews, larger and more complex offsetting projects are associated with increased oversight and better potential for long-term benefits due to larger investments, available resources, and planning, as well as increased resilience [8, 18, 24, 39, 43]. However, such studies have not been conducted extensively for banks, especially those targeting freshwater systems and species, which may have case study-specific application differences. While debit and credit generation and on-the-ground application have

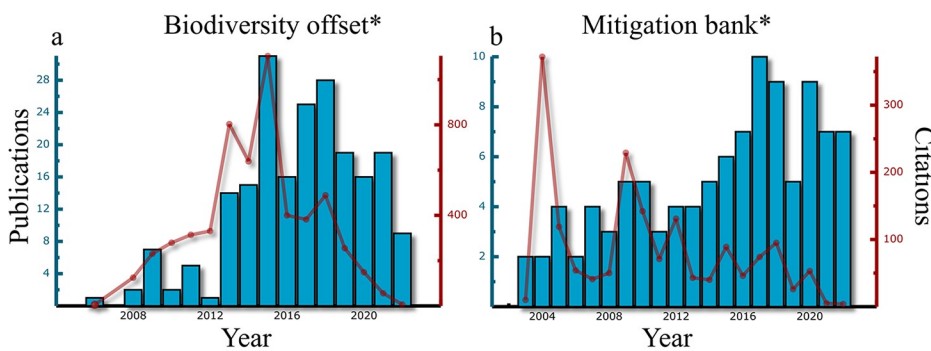

Commonly covered topics in studies and articles

**'How'?** - specific bank type or application review

**'Why and how'?** - framework reviews

**'Why and what'?** - framework and outcome review

**'What'?** - ecological outcome

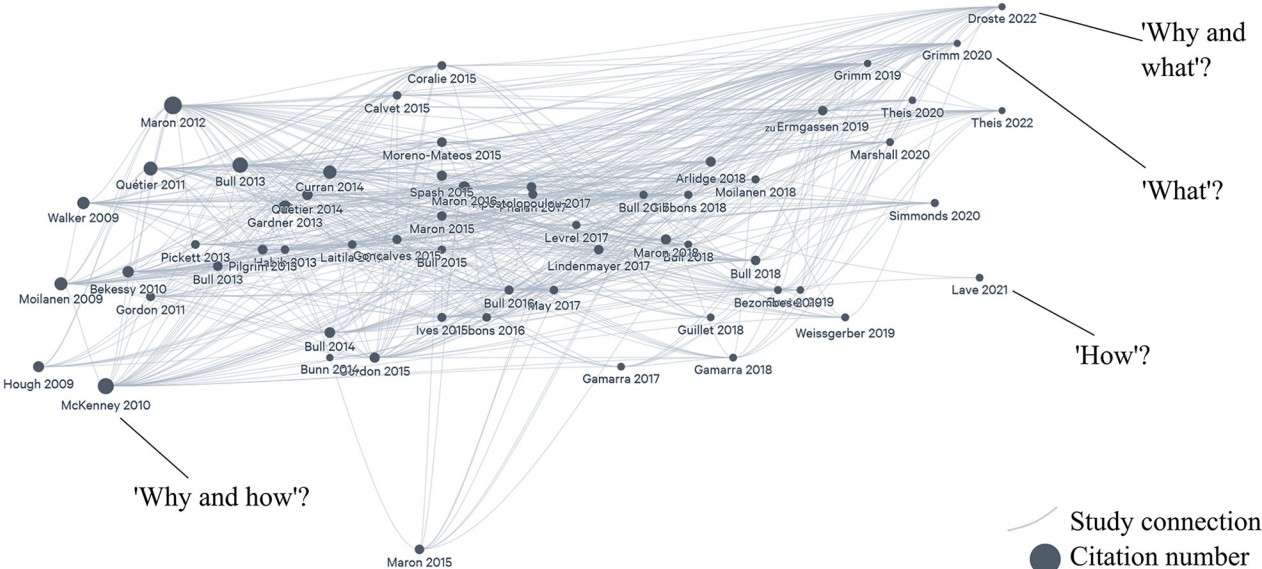

**Fig 2. Scientific uptake on offsetting and banking.** Increase in scientific publications on biodiversity offsetting (a) and mitigation banking (b) over the past 20 years (2002–2022) based on Google Scholar. *Indicate the use of variations of the terms offset and banking. Most connected and comprehensive studies and reviews (n = 59) on offsetting between 2002 and 2022, covering aspects of the 'Why' are we offsetting, 'How' are we offsetting, and 'What' are we achieving, with key examples highlighted [41].

been discussed in the broader offsetting context, monitoring and credit release schedule considerations have received less in-depth analysis in the literature, as they are typically assumed to be adequately covered in the reviewed and approved banking agreement and enforced by regulatory agencies [1, 15, 42].

Given the reviewed literature and to address this gap, this study reviews banking case studies from the United States, specifically as part of the Regulatory In-lieu Fee and Bank Information Tracking System (RIBITS), to gather as much information as possible to better inform banking practices, given its increasing popularity and demand, especially for freshwater systems.

1. We hypothesize that control mechanisms aimed at freshwater fish species and habitats in the United States will increase in scale as banks and projects become more complex, as suggested by previous findings in the literature on offsetting projects.

Our objective is to showcase commonly used performance and assessment criteria for banks addressing fish habitat in the United States, and to explore their application in different offsetting situations and management scenarios, as well as potential caveats in said scenarios.

## 2. Methods

### 2.1 Data acquisition

We utilized the Regulatory In-lieu fee and Bank Information Tracking System (RIBITS) to obtain relevant banking documents. The slowraker package in R was used to conduct a key-word search [44], resulting in 208 matches out of 24,411 files on RIBITS. Manual screening of the files revealed 53 banks, of which 47 had sufficient data for our review, specifically focusing on fish habitat valuation, release schedules, and monitoring requirements. The reviewed banking cases encompassed a range of documents, from permit copies to detailed project construction plans and monitoring reports. Our search and information review were conducted between February 6th and April 6th, 2022 (S1 and S2 Figs). Key variables were derived from the 47 included banks and based on issues and uncertainties commonly associated with offsetting, as identified in the literature.

### 2.2 Bank parameters: Bank control mechanisms and bank complexity metrics

The main objective of this study was to test if bank control mechanisms scale with increasing bank complexity. The two main bank control mechanisms chosen were credit release schedules and monitoring timeframes (Table 1). Monitoring timeframes, the minimum timeframe a bank is required to monitor its sites was measured in years, based on the individual bank files (variable name: *Monitoring timeframe*). Credit release schedules synthesized from the individual bank files cover over what period in years a bank can release their credits to be sold to proponents (variable name: *Release schedule*; Table 1). These two aspects constitute our control mechanisms.

Bank complexity was based on five different metrics, summed in a single overall complexity score (variable name: *Bank complexity*). Complexity metrics were based on the individual bank files and aimed to cover biological as well as operational aspects (Table 1). The metrics chosen were Compensation method, Project performance criteria, Species targeted, Habitat targeted, and Credit/ debit calculation method. Each metric information was assigned to simple categorical scores to allow for a comparison between qualitative and quantitative data and make use of information that is normally deemed unsuited for empirical modeling [45, 46]. For instance, a bank targeting a single habitat aspect was assigned a score of 1 for habitat targeted, a 2 when targeting multiple aspects, and a 3 when targeting whole ecosystems. This was done for each metric and captured increasing complexity based on bank targets and operations (Table 1). Scores for each of the five metrics per bank were summed to form the overall bank complexity score (Example in Table 1).

Overall, this approach allows us to have a single score for bank complexity based on many different complexity aspects. Multi-metric indices are applied commonly in environmental studies and hold the advantage of being able to collapse complex systems or features into tangible indicators, that make it easier to inform management decisions or derive ecological insights into complex systems [47–50].

### 2.3 Bank parameters: Compensation approach–bank type

Banks can vary by default depending on their targets and management approaches. For example, a bank restoring connectivity via weir removal is expected to be less complex in terms of

**Table 1. Synthesized bank parameters translated to quality control mechanisms (monitoring timeframe and release schedule) and bank complexity (aggregated multi-metric index) as well as definition and use within the study (based on synthesis n = 47 banks).**

| Assessed banks within the United States | n = 47 | | |
|---|---|---|---|
| **Bank parameters** | **Definition** | **Final variable** | **Metric** |
| **Quality control mechanisms** | | | |
| Monitoring time frame or period | What are the officially required monitoring timeframes for the bank? | *Monitoring timeframe* | Years |
| Credit release schedule | Over what period are credits released throughout bank approval, establishment, and long-term operation? | *Release schedule* | Years |

Use: What are some **quality control mechanisms** that are used by banks to ensure net benefits?
Example:
Bank releasing 100% of their generated credits over 5 years (5) and having monitoring commitments for 10 years (10).
Caveat: There are more control mechanisms available as stipulated in most banking agreements but our focus here is on the two main ones. Furthermore, control mechanisms are seldom covered in literature studies with a heavy focus on ecological and environmental outcomes.

| Bank complexity metrics | Definition | Final variable | Metric |
|---|---|---|---|
| Compensation method | A bank can use enhancement/ preservation (least desired), restoration/ rehabilitation, and habitat creation/ re-establishment (most desired) as the main approach to achieve net benefits. Which approach is the main approach used by the individual bank? | *Compensation method* | Categorical scores based on commonly accepted value of method:<br>Enhancement/ preservation: 1<br>Restoration/ rehabilitation: 2<br>Habitat creation/ re-establishment: 3 |
| Project performance criteria | How many different performance metrics is a bank using to evaluate net benefits? | *Project performance and assessment metrics* | Categorical scores based on number of metrics:<br>1–3 metrics: 1<br>4–6 metrics: 2<br>>7 metrics: 3 |
| Species targeted by the bank | How many different species are targeted by the bank and its net benefits? Vulnerable or endangered species are commonly regarded as higher-value targets for banks. | *Number of species targeted* | Categorical scores based on number of species:<br>Single: 1<br>Multiple: 2<br>Multiple + endangered: 3 |
| Habitat targeted by the bank | Is a bank targeting a single ecosystem aspect with their compensation method, multiple ones, or an ecosystem as a whole? Holistic approaches are commonly deemed of higher value. | *Habitat targeted* | Categorical scores based on habitat targets:<br>Single aspect: 1<br>Multiple aspects: 2<br>Whole system/ habitat: 3 |
| Credit/ debit calculation method | Which measure is used to translate provided net benefits by the bank to sellable credits (Ratio, Condition, Function)? | *Credit/ debit calculation method* | Categorical scores based on credit/ debit methods:<br>Ratio method: 1<br>Condition: 2<br>Function: 3 |
| **Explanation for credit calculation methods:** | **There are three main approaches to how enhanced, restored, or created habitat is translated to credits that can be sold that range from simple to complex.** | | |
| | **Ratio method:** These methods are based on aerial or linear measures derived from a specific habitat type at the impact site compared to the offset site. Ratios do not assess functional gains and losses compared to reference sites or starting conditions. Ratio methods predict gains and losses based on the chosen compensation method (e.g., restoration or enhancement). For instance, wetland restoration could receive a 2:1 area-to-credit ratio while enhancement could receive a 4:1 ratio due to a lower expected benefit potential. Ratios differ from project to project and often rely on professional judgment and the science-based benefit potential of different compensation measures in combination with different ecosystems or habitat types. | | |
| | **Condition method:** These methods quantify the ability of an aquatic ecosystem or resource to maintain a species composition (abundance, diversity, functional organization) that is comparable to reference systems in the region or commonly accepted reference values. Conditions are generally assessed on a scale of 0 to 1, 1 meaning the restored system is comparable to the chosen reference site. Condition scores allow for comparison among impact sites, offset sites, and reference sites. | | |
| | **Function method:** Degree to which an aquatic habitat is performing its intended or a specific function (e.g., is stream substrate providing spawning opportunities). Functional assessments can be of a physical, chemical, or biological nature. Assessing functions often requires temporal and spatial repetition and thus can be time-intensive and expensive. | | |

*(Continued)*

**Table 1.** (Continued)

| Assessed banks within the United States | n = 47 | | |
|---|---|---|---|
| **Bank parameters** | **Definition** | **Final variable** | **Metric** |
| **Quality control mechanisms** | | | |
| **Bank complexity score:** | **Bank complexity metrics** are summed for each bank.<br>Example:<br>Bank using enhancement (1) to benefit a single salmonid species (1) with main performance criteria being an increase in spawning success (1) through targeting a single habitat aspect (1; spawning gravel addition) with credits being calculated through a change in stream condition score (2).<br>Total complexity score of 6. | | |

Use: Do **Quality control mechanisms** scale with increasing **Bank complexity?**
Caveat: There are potential bank differences by default depending on their targets and approach that could be reflected in complexity and control mechanisms, hence the inclusion of bank type based on the compensation approach (Table 2).

monitoring and management approaches as opposed to a bank creating a new pond and wetland complex from the ground up [34, 51–54]. Hence it is vital to account for these differences by including bank type based on different compensation approaches in the analyses (variable name: *Compensation approach*; Table 2). Based on the individual bank files (n = 47) and supported by literature [15, 18, 20, 52], there are three main bank types based on compensation approaches.

We classified banks that addressed connectivity aspects through barrier removal or connectivity restoration as Connectivity and Barrier Removal banks (CBR; n = 22; Table 2). Restoring connectivity between upstream and downstream habitats or pond and floodplain habitats is essential for conservation efforts targeting anadromous fish species, such as Alewife (*Alosa pseudoharengus*), Atlantic Salmon (*Salmo salar*), American Eel (*Anguilla rostrata*) and Brook Trout (*Salvelinus fontinalis*). Barrier removal is the most common compensation method used for credit generation that addresses connectivity issues [52, 55–57].

Banks that address physical aspects such as geomorphology, bank erosion, or changes in habitat condition based on habitat type or class are classified as Physical Aspects and Class (PAC) banks (n = 14; Table 2). Improving key physical aspects of degraded or impaired habitat is a common approach to generating bank credits [1, 15, 18, 20, 58, 59]. PAC bank projects, with a special focus on wetland fish habitat, use habitat classes and changes in condition metrics measured through well-defined performance criteria, taking into consideration area and proximity, to allow large-scale projects with a transparent debit-to-credit translation process and NNL as the overall goal [1, 15, 18, 60].

Habitat, Community, and Species banks (HCS; n = 11; Table 2) aim to generate credits by addressing fish habitat, species-specific habitat, or entire ecosystems. This approach often involves a combination of other strategies and landscape planning processes to improve fish abundance, biomass, productivity, and spawning success. These banks establish the goal of restoring crucial habitat features such as hydrology or connectivity, creating new aquatic habitat areas, increasing habitat function and complexity, and benefiting aquatic-dependent wildlife or specific target species. HCS banks often consider other land-use aspects such as agriculture and logging, aiming to restore sites to pre-impact conditions on a landscape level, exceeding individual impacts [12, 13, 16, 18, 32, 53, 61].

Including bank type into our bank complexity vs. control mechanism analyses allows us to not only control for inherent differences but also make potential inferences on general management recommendations and caveats and application scenarios for each type.

**Table 2. Synthesized bank compensation approaches translated to distinct bank types and commonly used project performance and assessment metrics as well as definition and use within the study (based on synthesis n = 47 banks).**

| Bank parameters | Definition | Final variable | Number from total sample size n = 47 |
|---|---|---|---|
| **Compensation approach–bank type** | What ecological compensation approach is the bank using to provide environmental and ecological net benefits? | *Bank type* | |
| Connectivity and Barrier Removal banks | Banks benefit a system or species through barrier removal or connectivity restoration. Often uses ratio methods of benefiting habitat (e.g., upstream, and downstream connectivity). | *CBR* | n = 22 |
| Physical Aspects and Class | Banks benefit a system or species by targeting key physical aspects of degraded or impaired habitats. Use habitat classes and changes in condition metrics. | *PAC* | n = 14 |
| Habitat, Community, and Species | Banks benefit a system or species by addressing fish habitat, species-specific habitat, or entire ecosystems. This approach often involves a combination of other strategies and landscape planning processes (holistic). | *HCS* | n = 11 |

Use: Do **Quality control mechanisms** scale with increasing **Bank complexity** across the three different **Bank types?**

Caveat: There are bank differences in the way bank area is managed and assessed that are not part of the analyses (e.g., original performance metrics and criteria not translated to general scores) but could help understand application scenarios of the different bank types hence the inclusion of summary statistics for:

| Project performance and assessment metrics (detailed) | Definition | Final variable |
|---|---|---|
| Hydro/geomorphology | If and how is benefiting hydrology at bank sites assessed and monitored? | *Erosion and bank stability* *Flow* *Hydrological aspects (other than flow)* |
| Protection and control | If and how are bank sites managed and protected? | *Invasive species control* *Maintenance and access restriction* |
| Fish populations | If and how are benefiting fish populations assessed and monitored? | *Abundance* *Biomass* *Presence absence* *Spawning success* *Diet* *Community structure* |
| Riparian buffer | If and how is benefiting riparian area and buffer assessed and monitored? | *% vegetation cover* *Stem count of planted vegetation* *% survival of planted vegetation* *% shaded area* |
| Biochemical | Are water and soil quality at bank sites monitored and assessed? | *Water quality* *Soil samples* |
| Habitat features | If restored, created, or enhanced, are habitat features like spawning gravel or woody habitat assessed and monitored at bank sites? | *% aquatic cover* *Spawning and rearing habitat* *% structural integrity of restoration and enhancement measures* *Substrate type and dominance* |

Use: Helps characterize the three bank types based on the way they use **Project performance and assessment metrics.** The listed metrics are used in terms of **the frequency of occurrence of each project performance and assessment metric** across the three **bank types.**

| Additional bank considerations | Definition | Final variable | Metric |
|---|---|---|---|
| Credit release schedule | How many credits (in percent) are released at what point in time throughout bank approval, establishment, and long-term operation? | *% credits released* | % of credits released over time |

Use: Helps characterize the three bank types based on the way they commonly release and make generated credits available over time in percent of total possible credits.

## 2.4 Bank parameters: Project performance and assessment metrics

The mentioned potential management recommendations, caveats, and application scenarios for each bank type were rounded out by recording the frequency of project performance and assessment metrics used in each bank. Project performance and assessment metrics recorded were Hydro/geomorphology, Protection and control, Fish populations, Riparian buffer,

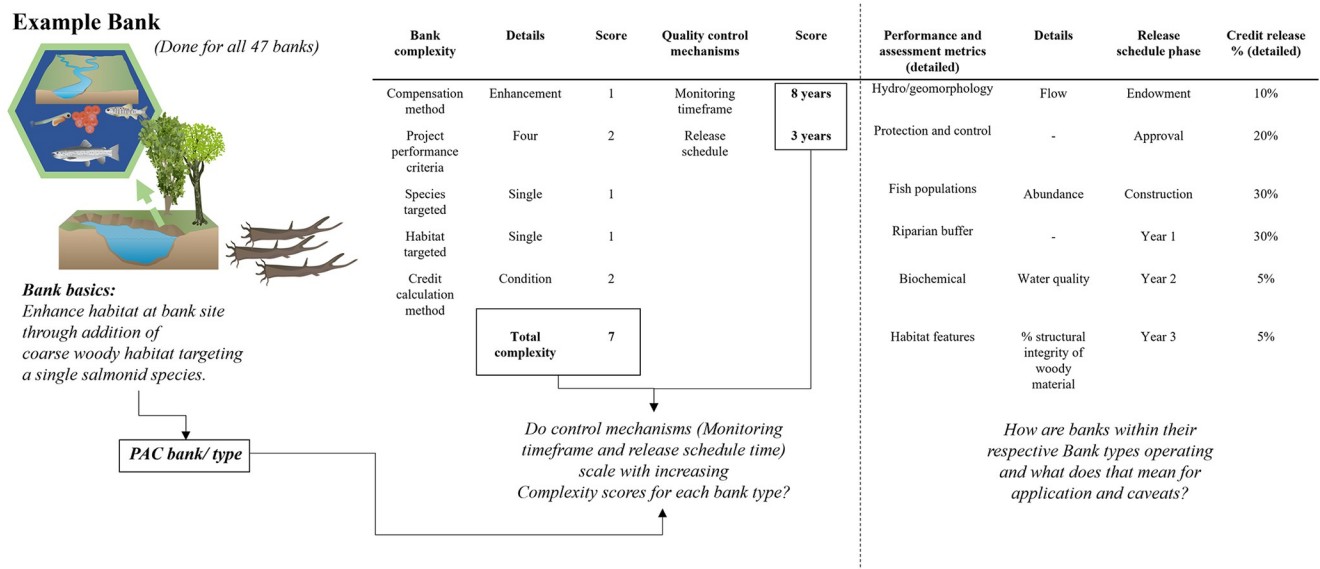

**Fig 3. Bank scores example.** Example bank showcasing how complexity and control mechanisms scores were formed for all 47 banks used to test if control mechanisms (Monitoring timeframe and release schedule time) scale with increasing Complexity scores for each bank type. Additional information is used to understand how banks within their respective bank types operate and what it could mean for applications and caveats (image attribution: [11]).

Biochemical, and Habitat features (based on synthesized bank files, n = 47; Table 2) with respective individual aspects (variables). For instance, the metric of Protection and control covers two individual aspects (variables), Invasive species control and Maintenance and access restriction to bank sites. Another example is how fish populations were assessed by banks. The metric of assessing fish populations covered six commonly used aspects ranging from Abundance to Diet composition (Table 2).

Finally specific information regarding credit release schedules, aside from the initially mentioned time in years, which is part of bank complexity vs. control mechanisms, was collected. The focus here was on what percentage of credits was released in what year (e.g., 10% of total supply released after 1 year). Information on percent credit release over time, like the above project performance and assessment metrics, can help put results from bank complexity vs. control mechanisms under consideration of the three different bank types into context.

Project performance and assessment metrics as well as credit resale in percent over time across bank types are essential puzzle pieces helping to characterize the three bank types based on the way they operate which will help understand potential results from the complexity question. Fig 3 is an example meant to illustrate the different goals, and variables and how they complement each other to create a more holistic picture of bank operation based on an example bank (Fig 3).

## 2.5 Analyses

**2.5.1 Bank control mechanisms with increasing bank complexity.** Monitoring timeframe and Release schedule (Table 1) were the main response variables, responding to increasing Complexity scores (Table 1) within a Generalized Additive Modeling framework (GAM). Six models were used one for each Bank type (n = 3) and Monitoring timeframe and Release schedule vs. Bank complexity combination: Bank type (CBR, PAC, HCS): Release schedule (or Monitoring timeframe) ← Complexity score. GAMs do not assume underlying linear

relationships between response and predictor and hence are better suited over for instance a linear regression when potential relationships are unknown [62, 63]. They offer a good trade-off between flexibility and interpretability situated between simple linear models and machine learning as well as are common in ecology where datasets often do not meet assumptions of normality or heteroscedasticity [62, 63].

GAMs use smooth terms which are functions of a wide variety of shapes that help fit the data [63]. GAMs were validated by using a k-index>1, a commonly accepted benchmark when fitting the model. k' controls the wiggliness (number of basis functions used to build a smooth function) of the smooth term that fits the data. The k-index captures the relationship between effective degrees of freedom edf (degree of non-linearity of a curve) and k'. k indices >1 indicated edf smaller than k' and thus a model fits the data better and is not overly constrained [63, 64]. Overall GAMs in our case are well suited since they allow the investigation of whether higher bank complexity is associated with longer release schedules and monitoring requirements, as well as identifying the form of relationship between project complexity, monitoring, and release schedule timelines even when non-linear ([63, 64]; R-4.2.2).

**2.5.2 Bank operation based on performance and assessment criteria frequency.** Performance and assessment aspects under the six broad categories of Hydro/Geomorphology, Protection and control, Fish populations, Riparian buffer, Biochemical and Habitat features (Table 2) were summarized by proportionate frequency across the three bank types (CBR, PAC, HCS). These summary statistics and how often each performance and assessment criteria are used across bank types are meant to allow a better understanding of bank operation across types.

**2.5.3 Bank operation based on % credit release over time.** Percent of credits released (%) over bank stages (Stage 1 = Procurement–endowment and approval; Stage 2 = Construction; Stage 3 = Performance and management–Year 1 to 3; Stage 4 = Long-term performance and management–Year 4 to x) was done through Analyses of variance (ANOVA) as well as under consideration of the three bank types (CBR, PAC and HCS), followed by pairwise comparisons (Bonferroni corrected p-values; accepted alpha $< 0.05$; R-4.2.2; [65]). Keeping track of what proportion of credits are released over what period across bank types further provides insights into bank operation and if potential patterns (e.g. fast or slow release for a specific bank type) hold management implications or caveats [42].

## 3. Results

### 3.1 Project (bank) performance and assessment metrics

The three primary bank types place varying emphasis on ecosystem aspects that are commonly accepted for quantifying the benefits of habitat or ecosystem function, condition, and species metrics during estimation and monitoring. For example, they differ in their emphasis on ecosystem aspects for quantifying habitat or ecosystem function, condition, and species metrics when estimating and monitoring benefits (S1 Table).

Connectivity and Barrier Removal projects mainly focused on Hydro/Geomorphology, with flow and hydrology aspects measured in all 22 assessed banks and 68.1% assessed bank erosion and stability. Species presence (fish; 100%) and maintenance and access requirements (protection and control; 100%) were other key aspects. Riparian buffer, biochemical aspects, and habitat features were less commonly considered (all $< 25\%$).

Physical Aspects and Class banks, like Connectivity and Barrier Removal types, often used Hydro/Geomorphology as assessment and performance metrics (Erosion and stability 50%; Flow 78.6%; Hydrology 64.3%). Physical Aspects and Class banks mainly relied on species presence (50%) and abundance (35.7%) to assess effects on fish populations. In comparison to

Connectivity and Barrier Removal banks, Physical Aspects and Class banks utilized Riparian buffer metrics like vegetation survival (50%) and overall cover (35.7%), as well as biochemical aspects like water quality (50%) and habitat features like aquatic cover, spawning habitat, structural integrity of habitat features, and substrate type (all ~ 30%; S1 Table).

Habitat, Community, and Species as the last of the three bank types utilized all listed assessment and performance metrics with a strong focus on hydrology (100%), invasive species control (81.8%), species abundance and presence (both 81.8%), vegetation survival (72.7%), water quality (72.7%), and aquatic cover (63.6%). Habitat, Community, and Species (HCS) banks also employed criteria that were less commonly or not used at all by the other bank types, such as fish diet, community structure, and soil sampling (S1 Table).

## 3.2 Bank complexity across release schedules and monitoring timeframes

**3.2.1 Bank complexity across release schedules and monitoring timeframes for all bank types.** The average complexity score for all 47 banks investigated was 9.49 (±2.01), and the average monitoring timeframe was 9.09 (±3.66) years. The average release schedule time was 5.19 (±2.13) years.

**3.2.2 Bank complexity across release schedules and monitoring timeframes for different bank types.** Connectivity and Barrier Removal banks (n = 22) had a mean complexity of 8.41 (±1.26) and a mean monitoring timeframe of 7.00 (±1.77) years, with an average time of 3.72 (±1.20) years to release all credits. There was no significant relationship between high bank complexity and increasing monitoring timeframes (p = 0.442; k-index = 1.02) or release schedules (p = 0.940; k-index = 1.14; Fig 4A and 4D).

Physical Aspects and Class banks (n = 14) showed an average complexity of 9.36 (±1.65), with an average monitoring timeframe of 9.79 (±3.85) years and 5.79 (±1.85) years for release schedules. There was no strong relationship between bank complexity and monitoring timeframes (p = 0.27; k-index = 1.28), but release schedule requirements in years increased with higher complexity Physical Aspects and Class banks (p = 0.038; k-index = 1.27; Fig 4B and 4E).

Habitat, Community, and Species banks (n = 11) had the highest mean complexity (11.8 ±1.78), as well as monitoring (12.36±3.72 years) and release schedule timeframes (7.36±1.75 years). Monitoring timeframes were only weakly related to increased complexity (p = 0.086; k-index = 1.47), while release schedule requirements in years increased with higher bank complexity (p = 0.036; k-index = 1.43; Fig 4 and 4C, 4f; full output tables can be found in the supplemental section S2–S4 Tables).

## 3.3 Release schedule across bank stages and bank types

**3.3.1 Credit release across stages.** The proportion of credit release as a percentage varied significantly across the four identified stages (df: 3; F-value: 31.83; p-value < 0.001; Fig 5A) as well as between the three bank types (df: 6; F-value: 10.71; p-value < 0.001; Fig 5B). During stages 1, 2, and 4, banks released approximately 20% of their total credit supply. Stage 3, which covered years 1 to 3 post-construction, had the highest average credit release of 39.6% (±14.9), significantly higher than the other three stages (p-value < 0.001).

**3.3.2 Credit release across stages and bank types.** The proportion of credit release in percent varied across the four stages (df: 3; F-value: 31.83; p-value < 0.001) and between the three bank types (df: 6; F-value: 10.71; p-value < 0.001). During stages 1, 2, and 4, banks released approximately 20% of their total credit supply. Stage 3, which covered years 1 to 3 post-construction, had the highest average credit release with 39.6% (±14.9), significantly higher than the other three stages (p-value < 0.001).

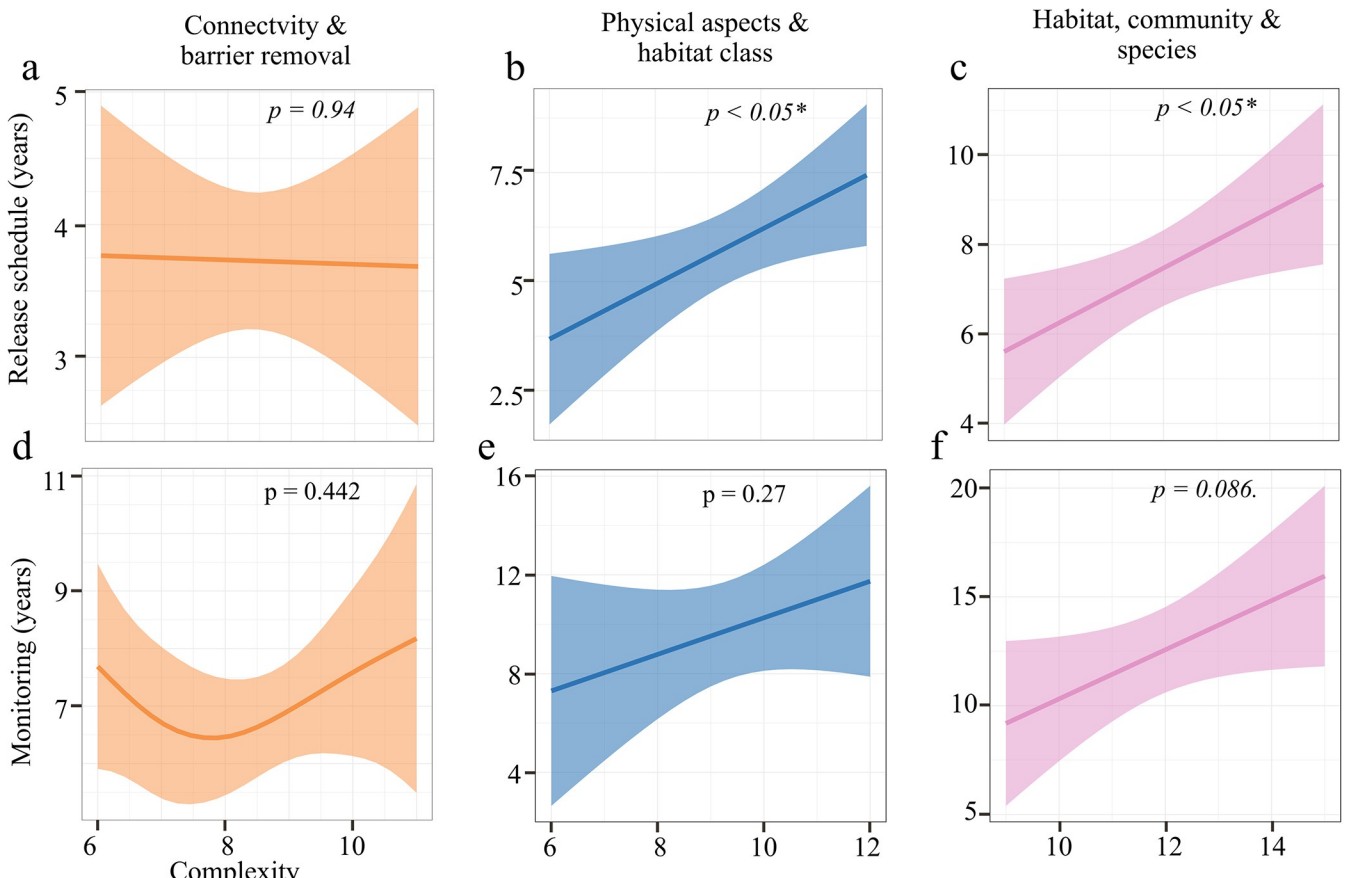

**Fig 4. Monitoring and credit release in relation to bank complexity.** Relationship between monitoring (years) and release schedule (years) and bank complexity for the three main bank types Connectivity and Barrier Removal banks (a, d, n = 22), Physical Aspects and Class banks (b, e, n = 14), Habitat, Community, and Species banks (c, f, n = 11). Outputs based on Generalized Additive Models (GAMs).

In stage 1, credit release did not differ between Habitat, Community, Species, and Physical Aspects and Class banks (14 to 16% of total supply) but was significantly higher for Connectivity and Barrier Removal banks compared to the other two (23.6%±14.7; p-value < 0.05; Fig 5C). Credit releases in stage 2 were comparable for Physical Aspects and Class and Connectivity and Barrier Removal banks with 23.1% (±11.4) and 20.4% (±9.1), respectively. Connectivity and Barrier Removal credit releases for stage 2 were significantly higher compared to Habitat, Community, and Species banks (p < 0.05; 14.6%±6.5).

Similar results were observed for stage 3, with Connectivity and Barrier Removal banks releasing on average 12% more (43.5%±17.9) than Habitat, Community, and Species banks (31.8%±8.2; p-value < 0.05). Finally, all bank types differed significantly (p < 0.05) in stage 4 regarding their proportionate credit releases. Connectivity and Barrier Removal banks released 9.8% (±1.2) of their credits, Physical Aspects, and Class banks released 23.9% (±14.7) of their credits, and Habitat, Community, and Species banks released 39.6% (±7.2) of their total credit supply.

Overall, Connectivity and Barrier Removal banks released credits faster, with a greater focus on stages 1 and 2 compared to Physical Aspects and Class and Habitat, Community, and Species banks. Physical Aspects and Class banks retained credits for stage 4, releasing fewer credits in stage 1 compared to Connectivity and Barrier Removal banks, and most credits for

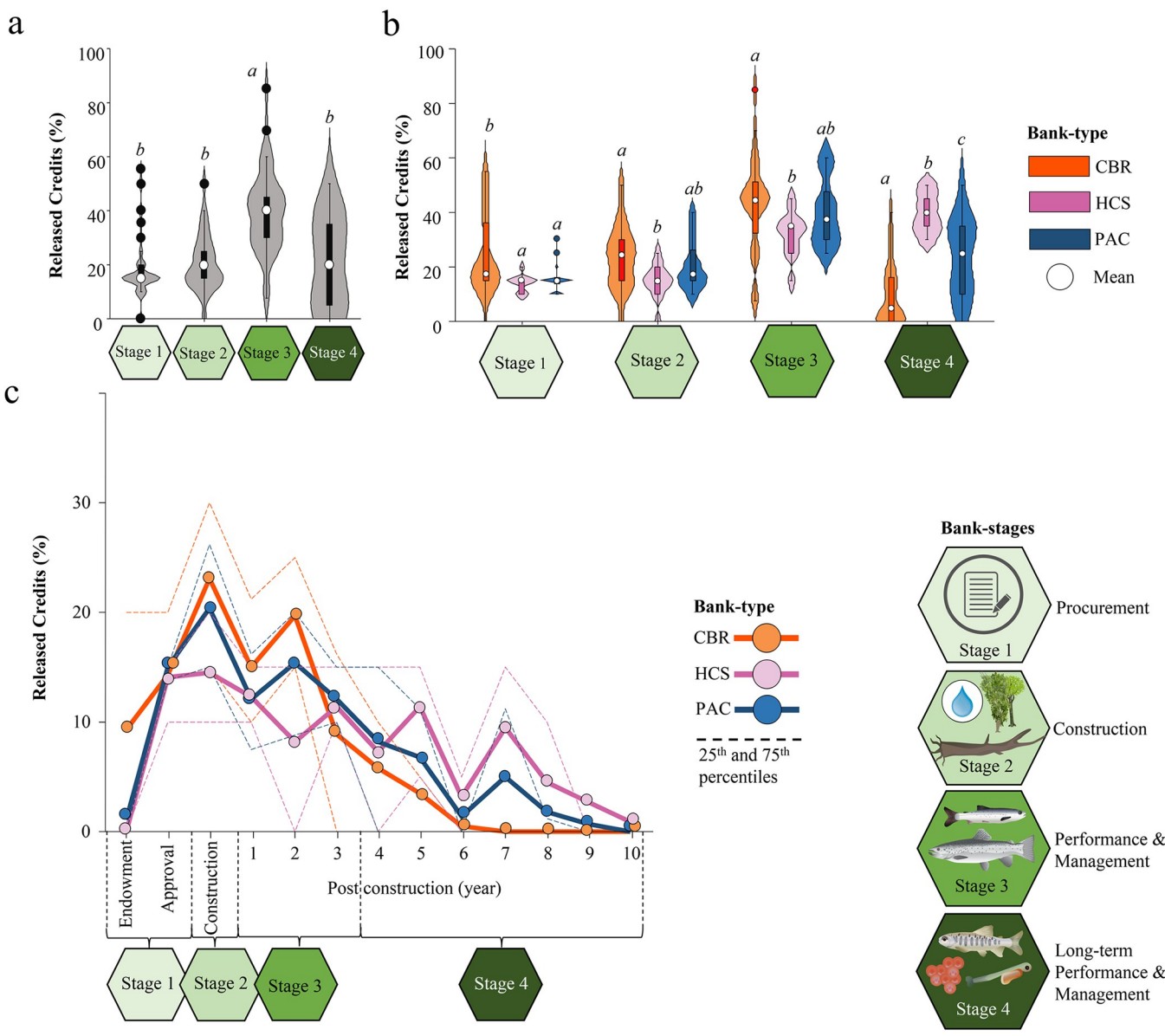

**Fig 5. Credit release amounts over time and bank type.** Credit release in percent (%) over different bank stages (a) (procurement, construction, (long-term) performance and management) and across different bank types (b) (Connectivity and Barrier Removal banks (n = 22), Physical Aspects and Class banks (n = 14), Habitat, Community, and Species banks (n = 11). Significant differences are indicated by letters (a, b, c). Detailed year-by-year comparison in panel c (image attribution: [53]).

Habitat, Community, and Species banks were released during stages 3 and 4 (Fig 5C; full output tables can be found in the supplemental section S5 Table).

## 4. Discussion

Our study shows that banks focusing on freshwater species and systems tend to adjust their release schedules and monitoring timeframes as their bank complexity increases. However, when examining the three bank types separately, we found some operational differences. The key findings can be summarized as follows:

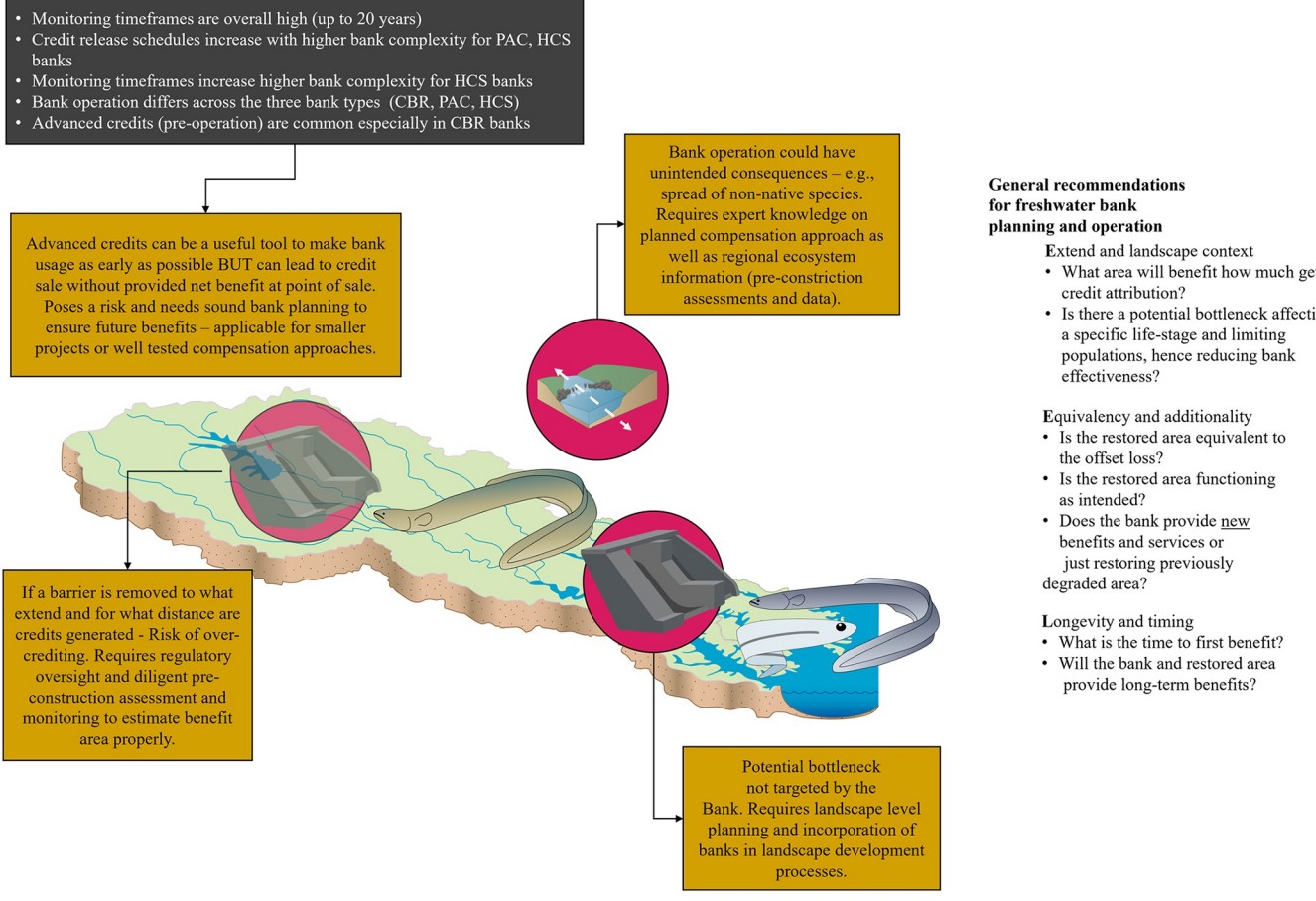

**Fig 6. General findings and caveats.** General findings from the synthesized freshwater banking data regarding relationships between bank complexity and control mechanisms as well as bank types and application scenarios as well as caveats with high-level recommendations for bank planning and operation (image attribution: [11]).

- Monitoring timeframes are generally high, often ranging from 10 to 20 years.

- Most credits are released during the approval and construction phase and the first 3 years post-restoration.

- Banks that focus on connectivity and barrier removal have lower overall complexity and seldom adjust monitoring timeframes and release schedules based on complexity.

- Banks that target habitat and species often release credits over a longer period, linked to meeting long-term performance criteria of up to 10 years post-construction.

These results provide general insights into whether monitoring and release schedules scale with complexity and offer different management applications for each bank type (Fig 6). These findings are particularly relevant when considering the listed performance and assessment metrics in Tables 1 and 2.

## 4.1 Complexity and release schedule/ monitoring

Our findings are encouraging, given the long monitoring timeframes and staggered release of credits over time, particularly for early and long-term ecological benefits during stages 3 and 4.

In contrast, offsetting projects often suffer from inadequate monitoring timeframes, insufficient enforcement of performance criteria, and a lack of assurance of long-term benefits [7, 8, 22, 66].

Our results for freshwater banks suggest that control mechanisms are generally adequately scaled and implemented, which can be an advantage, considering the increased demand for freshwater banks on a regional level across North America and the desire for municipalities to adopt banking systems into their development plans [14, 16, 18, 67].

## 4.2 Application

Connectivity and Barrier Removal banks have the potential to provide significant ecosystem benefits by restoring connectivity and removing barriers that limit fish access to rearing and spawning grounds. Loss of connectivity has a significant impact on the distribution of freshwater fish, particularly migratory species, and can affect cool and cold-water species' access to climate refugia [68–71]. Connectivity and Barrier Removal banks have lower complexity and monitoring times compared to the other two bank types, as removing smaller dams or weirs is often seen as a feasible measure with fast turnaround times and quick results [60, 68, 69]. Monitoring and assessments often rely on presence-absence measurements to confirm the restored passing ability of certain fish species, which is time and cost-efficient. Connectivity and Barrier Removal banks' lower complexity and faster turnaround times make them an attractive option for providing early and long-term ecological benefits [20, 60, 68].

Physical Aspects and Class banking projects differ from Connectivity and Barrier Removal projects in that they apply additional on-ground restoration and rehabilitation measures. They focus on promoting natural water storage through meadow, stream, and forest restoration, reducing erosion and sedimentation, and protecting and restoring floodplain and off-channel connectivity. Structural changes and additions to fish habitat, such as spawning substrate or large woody debris, are often used to achieve net benefits. The scale and complexity of Physical Aspects and Class projects can vary which allows for flexibility [53]. Transferable credits that are translatable between impacts and projects are widely used and can enhance flexibility even further. The main application of Physical Aspects and Class projects seems to be compensating for impacts related to road construction, forestry, and agriculture.

Physical Aspects and Class projects rely on sound scientific knowledge for indicator development, habitat classification, and adjustment factor validation [33, 53]. The projects assessed were mostly transparent in their approach, allowing for tailoring to individual stakeholder needs, including social values, number of indicators, and proximity and location factors. Transparency and the ability to scale effort up and down are essential for feasibility, particularly for small-scale or specialized projects. Our assessed projects show that Physical Aspects and Class banks generally require long-term protection measures and a structured release schedule to ensure long-term success and meet performance criteria.

Bank establishment targeting whole aquatic ecosystems (Habitat, Community, and Species) aims to restore essential habitat features like hydrology or connectivity while simultaneously creating new aquatic habitat areas, increasing habitat function and complexity, and benefiting aquatic-dependent wildlife or specific target species. Habitat-focused banks often take other land-use aspects into account, actively aiming to restore sites to pre-impact conditions on a landscape level, exceeding individual impacts (Function and Condition method, Table 1; [72–75]).

Many Habitat, Community, and Species banks are established because of long-term environmental degradation resulting from urban development, agriculture, forestry, or environmental pollution [72–74]. These banks measure functional gains in fish habitat based on the Habitat Equivalency Analysis (HEA), which can be used for habitat benefits as well as

species benefits. HEA is most often used to evaluate the gain of ecological function for stream habitats like estuaries or floodplains and their use for anadromous species and to identify biodiversity hotspots [76, 77]. Habitat-focused approaches taking species life history, temporal aspects, community goals, and the role of specific habitat types into account are highly recommended to be used when establishing a debiting and crediting system that aims to value fish habitat holistically and comprehensively (Condition and Function method, Table 1; [13, 72–74, 76, 77]). Habitat, Community, and Species can be applied to larger urban areas and large-scale impacts like mining, leading to long-term habitat degradation or offset requirements that often cannot be tackled through mitigation steps or a single offset [13, 16, 74, 76, 78].

Habitat, Community, and Species projects and frameworks are extensive and require qualitative and quantitative data and baseline assessments as well as collaborative support and third-party management and control to avoid pitfalls (e.g., Interagency Review Team (IRT); Condition and Function method, Table 1; [16, 33, 79]). Long-term monitoring and staggered credit release over a long time aim to address increasing complexity and ensure long-term benefits [19, 42]. Consequently, Habitat, Community, and Species approaches may not be feasible for small-scale projects or projects not part of a larger planning or management framework or service area [74, 80].

Given the different nature of bank types described in our study, certain aspects and caveats need to be considered when implementing freshwater banks, relating to both regulatory as well as management-related issues.

## 4.3 Fast credit release and advance credits

Our study indicates that many credits in the assessed banks were released within the first three years, including during the approval and construction phase. However, this fast credit release may not align with the timelines needed for ecological benefits to be realized (Fig 6). Previous studies have also shown this to be a concern [10, 14, 16, 37, 66]. One extreme example of the fast provision of ecological currency is the use of advance credits. Our analysis revealed that a high proportion of advance credits were granted and released before project construction, particularly for Connectivity and Barrier Removal banks, which have an expected fast turnaround time for ecological benefits. However, this practice can result in sold credits before meeting any ecological or performance benchmarks, as noted by previous studies [21, 80] It is worth noting that around 25% of credits in the assessed Connectivity and Barrier Removal banks were offered pre-construction, which highlights a significant caveat to their application.

Our findings suggest that advance credits are more commonly used in Connectivity and Barrier Removal banks due to the focus on restoring connectivity, which is often considered a proven approach to achieving ecological benefits [42, 52]. However, this practice also carries uncertainties and requires long-term monitoring to ensure that in-perpetuity requirements are met [42, 53, 81]. Given the need for sustained monitoring, it may be appropriate to consider reducing the number of advance credits granted by Connectivity and Barrier Removal banks to achieve a more balanced approach.

Furthermore, banking agreements should ensure long-term funding, making advance credits unnecessary for generating immediate revenue [21, 23]. Therefore, regions with low credit demand and banks with sound financing models and business plans should avoid relying on advance credits. In contrast, Habitat, Community, and Species and Physical Aspects and Class banks, which are more complex and have fewer advance credits, could serve as alternative models for achieving ecological benefits in a more balanced manner.

### 4.4 Unintended consequences

Unintended consequences of mitigation and restoration techniques are often overlooked, but they pose a significant risk to offsetting projects [33, 79, 82]. For example, a mitigation method that benefits one species may have negative effects on another, making invasive species management and contingency planning essential [83]. Unfortunately, contingency plans for unintended consequences are rarely included in banking agreements. These issues are difficult to predict, but regular monitoring and strong performance criteria can mitigate risks and ensure positive ecological outcomes (Fig 6 [21, 22, 81]). More studies are needed to develop best practices for bank owners and proponents to prevent unintended consequences and manage them effectively.

### 4.5 Over-crediting

Finally, over-crediting is an often-underestimated issue that affects freshwater banks, especially those in riverine systems using a ratio methodology (Table 1). Over-crediting refers to the accreditation of too many credits relative to the actual implemented offset. A prime example is barrier removal, where restoring connectivity within a 50 km stretch of stream could be argued to receive credits for all 50 km. This example highlights the often-difficult process of estimating credits accurately (Fig 6 [19, 52, 81]). While certain control methods and adjustment factors attempt to estimate the area of provided benefit (e.g., North Carolina Guidance, New England District Method), there is still significant uncertainty [52, 59, 60].

A market-driven system like banking requires a balancing act between incentivizing proponents or landowners to invest while ensuring equivalent ecological benefits [14, 16, 21, 58]. This also applies to credit prices, allowed service areas, and providing proper regulatory and scientific support and resources [58, 81, 84].

Overall, credit estimation and credit types are among the most critical aspects of banking and require further in-depth studies to determine if current practices in the United States need improvement. Credit-focused studies would also benefit other countries in developing their crediting systems and avoiding issues and pitfalls present but mostly unevaluated in the United States.

## 5. Conclusions

Based on the results from our synthesis we show that bank monitoring timeframes for freshwater banks in the United States are generally high, often ranging from 10 to 20 years which is an encouraging finding since the environmental market often lacks oversight and long-term monitoring [7, 8, 79, 85]. Credits are released over multiple bank stages and years with most credits being released during the approval and construction phase and the first 3 years post-construction. Attention needs to be paid to advanced credits to avoid potential net loss if anticipated benefits are not provided [19, 42]. Advanced credits should be avoided for banking projects and should only be considered for well-studied compensation approaches and small feasible projects [19, 42, 52]. Banks that focus on connectivity and barrier removal have lower overall complexity given their popularity for often smaller barrier removal and rarely adjust monitoring timeframes and release schedules based on their lower complexity with a focus on the ability of fish to pass [52, 56]. What area of stream or river receives credits for barrier removal needs to be grounded in through pre-restoration assessments and monitoring to avoid over-crediting. Banks that target habitat and species often release credits over a longer period, linked to meeting long-term performance criteria of up to 10 years post-construction. These banks, especially, need to consider larger landscape contexts and benefit from being incorporated into landscape and development planning processes [12, 13, 26, 61, 72, 86].

Unintended consequences considerations are important for all bank types and restoration efforts in general, with bank management and operation potentially benefiting from knowledge-sharing networks and continuous logistics support through regulatory agencies [53, 55, 75, 82, 83].

## Supporting information

**S1 Table. Bank monitoring.** Performance and assessment aspects used by the three bank types.
(DOCX)

**S2 Table. CBR banks model output.** Generalized Additive Model output for bank type monitoring and release schedule timeframes over bank complexity for CBR banks.
(DOCX)

**S3 Table. PAC banks model output.** Generalized Additive Model output for bank type monitoring and release schedule timeframes over bank complexity for PAC banks.
(DOCX)

**S4 Table. HCS banks model output.** Generalized Additive Model output for bank type monitoring and release schedule timeframes over bank complexity for HCS banks.
(DOCX)

**S5 Table. ANOVA table.** ANOVA output for the number of credits (%) released over different bank stages and pairwise comparisons for bank types.
(DOCX)

**S1 Fig. Workflow.** Workflow ranging from data acquisition to processing and analysis.
(PNG)

**S2 Fig. Boolean search terms.**
(TIF)

## Acknowledgments

We want to acknowledge the role that open access data and scientific material plays in conducting research. Habitat banking data is freely available through RIBITS, and scientific symbols used to enhance figures were provided through ian.umces.edu under the Attribution-ShareAlike 4.0 International (CC BY-SA 4.0) agreement. We want to thank Richard Kavanagh and Jason Shpeley from Fisheries and Oceans for their constructive and invaluable feedback.

## Author Contributions

**Conceptualization:** Sebastian Theis, Mark Poesch.

**Formal analysis:** Sebastian Theis.

**Funding acquisition:** Mark Poesch.

**Investigation:** Sebastian Theis.

**Methodology:** Sebastian Theis, Mark Poesch.

**Project administration:** Sebastian Theis.

**Visualization:** Sebastian Theis.

**Writing – original draft:** Sebastian Theis.

**Writing – review & editing:** Sebastian Theis, Mark Poesch.

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
