## [Decision Letter · Decision Letter 0]

8 Sep 2023

PONE-D-23-15042Mitigation bank applications for freshwater systems: Control mechanisms, project complexity, and caveatsPLOS ONE

Dear Dr. Theis,

Thank you for submitting your manuscript to PLOS ONE. After careful consideration, we feel that it has merit but does not fully meet PLOS ONE’s publication criteria as it currently stands. Therefore, we invite you to submit a revised version of the manuscript that addresses the points raised during the review process.

We look forward to receiving your revised manuscript.

Kind regards,

Susmita Lahiri (Ganguly)

Academic Editor

PLOS ONE

Journal Requirements:

"Funding for this project was provided by Mitacs Cluster Accelerate (RES0027784) and Converge (RES0021639) grants to M.P. We want to further acknowledge the role that open access data and scientific material plays in conducting research. Habitat banking data is freely available through RIBITS, and scientific symbols used to enhance figures were provided through ian.umces.edu under the Attribution-ShareAlike 4.0 International (CC BY-SA 4.0) agreement.We want to thank Richard Kavanagh and Jason Shpeley from Fisheries and Oceans for their constructive and invaluable feedback."

Funding information should not appear in the Acknowledgments section or other areas of your manuscript. We will only publish funding information present in the Funding Statement section of the online submission form. 

"Funding for this project was provided by Mitacs Cluster Accelerate (RES0027784) and Converge (RES0021639) grants to M.P.

(https://www.mitacs.ca/en/programs/accelerate)

Reviewers' comments:

Reviewer's Responses to Questions

**Comments to the Author**

1. Is the manuscript technically sound, and do the data support the conclusions?

Reviewer #1: Partly

2. Has the statistical analysis been performed appropriately and rigorously? 

Reviewer #1: No

3. Have the authors made all data underlying the findings in their manuscript fully available?

Reviewer #1: Yes

4. Is the manuscript presented in an intelligible fashion and written in standard English?

Reviewer #1: No

5. Review Comments to the Author

Reviewer #1: Introduction

- Include literature gap and study’s contribution. Apart from giving a good background, the introduction must reveal the existing literature gap and consequently what your study contributes to. Also, please, kindly read the instructions for the authors to be abide by the requirements of the journal. You might not need to isolate the literature review as a separate section, it could be part of the introduction in different paragraphs.

Material and methods

- The authors need to present the analytical framework in detail supported by literature, define and describe the variables used in the study and explain the model employed.

- Try to be more specific and clearer of variables measurement!

Results and discussion

- Please read the author’s guide and appropriately develop this section.

Conclusion

- The conclusion should be empirically supported by the results and should give conclusion on aspects presented in the results section. Authors must bear in mind that conclusion is simply a ‘take home message’ based on the findings presented. Therefore, I strongly suggest that authors reconstruct the write-up in the conclusion and include policy recommendations.

6. PLOS authors have the option to publish the peer review history of their article (what does this mean?). If published, this will include your full peer review and any attached files.

Reviewer #1: **Yes: **Essiagnon John-Philippe Alavo

---

## [Author Response · Author response to Decision Letter 0]

17 Sep 2023

PONE-D-23-15042

Mitigation bank applications for freshwater systems: Control mechanisms, project complexity, and caveats

PLOS ONE

We would like to thank Reviewers for taking the time and effort necessary to review the manuscript. We sincerely appreciate all valuable comments and suggestions, which helped us to improve the quality of the manuscript.

Reviewers' comments:

Reviewer's Responses to Questions

Comments to the Author

1. Is the manuscript technically sound, and do the data support the conclusions?

Reviewer #1: Partly

Reply: Thank you for your comment. 

As a quick overview here are the main changes done to the manuscript to improve technical soundness:

1. Inclusion of extended literature review to highlight the knowledge gap better.

2. Simplified overview and explanation on how mitigation banking/ offsetting works so it should be more accessible for people outside of the field (E.g., Figure 1).

3. Extension and rework of the method section to explain variables better and provided a step-by-step walkthrough of each variable and its role (E.g., Table 1 & 2).

4. Walkthrough example for a bank (Figure 3).

5. Extension of provided method explanations e.g., better justification for GAMs etc. in the method section.

6. Conclusion section with main take home messages and high-level recommendations.

2. Has the statistical analysis been performed appropriately and rigorously? 

Reviewer #1: No

Reply: 

Thank you for your comment. Would you be able to specify which part of the analysis has not been performed appropriately or rigorously? There might have been some confusion based on the lengthy method section as well as combining variable descriptions and statistical approaches. Hence the section has been restructured to better reflect variable content and statistical measures separately. We also want to reiterate the use of supplements as e.g., Figure S1 holds much of the conceptual and analytical framework and should be used in conjunction with reading the manuscript. Please refer to the updated method section. If not here is a short summary on how things have been changed:

2.1 Still the same and covers basic data acquisition which should be intuitive.

2.2-2.4 Formerly key variables have been changed to two different full tables (1 & 2) that hold all qualitative and quantitative variables with description and use. This should address your concerns and help explain how they are used in their own respective way to answer the individual research questions. This also frees up space in the method section to justify and support variable use more. 

We also acknowledge that mitigation banking as well as offsetting is a still rather novel field with which not everyone is familiar in the environmental sector. Thus, we have added a new Figure 1 & 3 to the Introduction/ Methods to include a simplified approach on how mitigation banking works – a) is an example of traditional offsetting and b) the third-party introduction of a bank. Figure 3 is a walkthrough of calculating scores for an example bank.

2.5 Extension of used statistical analyses to explain and justify the use of GAMs better.

3. Have the authors made all data underlying the findings in their manuscript fully available?

Reviewer #1: Yes

Reply: -

4. Is the manuscript presented in an intelligible fashion and written in standard English?

Reviewer #1: No

Reply: 

Draft has been reviewed by native English speakers as well as grammar software. If you still feel that it is unclear, please specify the nature of your concerns in whether they refer to accessibility, comprehensiveness, or grammar/ spelling. Certain terms are inherently hard to describe outside the context of mitigation banking and might not be fully accessible or translatable to ley terms. 

5. Review Comments to the Author

Reviewer #1: Introduction

- Include literature gap and study’s contribution. Apart from giving a good background, the introduction must reveal the existing literature gap and consequently what your study contributes to. Also, please, kindly read the instructions for the authors to be abide by the requirements of the journal. You might not need to isolate the literature review as a separate section, it could be part of the introduction in different paragraphs.

Material and methods

- The authors need to present the analytical framework in detail supported by literature, define and describe the variables used in the study and explain the model employed.

- Try to be more specific and clearer of variables measurement!

Results and discussion

- Please read the author’s guide and appropriately develop this section.

Conclusion

- The conclusion should be empirically supported by the results and should give conclusion on aspects presented in the results section. Authors must bear in mind that conclusion is simply a ‘take home message’ based on the findings presented. Therefore, I strongly suggest that authors reconstruct the write-up in the conclusion and include policy recommendations.

We have extended the reviewed studies in the introduction to focus on the key available reviews covering offsetting and mitigation. We acknowledge that there are more studies out there, especially in a regional context but the mentioned ones (~50) are the keystone studies and papers that have attempted system wide and large-scale reviews and syntheses of the mitigation sector and framework.

Conclusions with take home messages have been synthesized from the discussion and added to the manuscript to increase accessibility.

---

## [Editor Report · Decision Letter 1]

27 Sep 2023

Mitigation bank applications for freshwater systems: Control mechanisms, project complexity, and caveats

PONE-D-23-15042R1

Dear Dr. 

,

We’re pleased to inform you that your manuscript has been judged scientifically suitable for publication and will be formally accepted for publication once it meets all outstanding technical requirements.

Kind regards,

Susmita Lahiri (Ganguly)

Academic Editor

PLOS ONE

---

## [Editor Report · Acceptance letter]

2 Oct 2023

PONE-D-23-15042R1 

Mitigation bank applications for freshwater systems: Control mechanisms, project complexity, and caveats 

Dear Dr. Theis:

I'm pleased to inform you that your manuscript has been deemed suitable for publication in PLOS ONE. Congratulations! Your manuscript is now with our production department. 

Kind regards, 

on behalf of

Dr. Susmita Lahiri (Ganguly) 

Academic Editor

PLOS ONE